# ChemSpacE: Interpretable and Interactive Chemical Space Exploration

**Yuanqi Du**                                                           *yd392@cornell.edu*
*Cornell University*

**Xian Liu**                                                     *alvinliu@ie.cuhk.edu.hk*
*The Chinese University of Hong Kong*

**Nilay Shah**                                                      *nshah76@g.ucla.edu*
*University of California Los Angeles*

**Shengchao Liu**                                                   *liusheng@mila.quebec*
*Mila, Université de Montréal*

**Jieyu Zhang**                                                *jieyuz2@cs.washington.edu*
*University of Washington*

**Bolei Zhou**                                                       *bolei@cs.ucla.edu*
*University of California Los Angeles*

**Reviewed on OpenReview:** *https://openreview.net/forum?id=C1Xl8dYCBn*

## Abstract

Discovering meaningful molecules in the vast combinatorial chemical space has been a long-standing challenge in many fields, from materials science to drug design. Recent progress in machine learning, especially with generative models, shows great promise for automated molecule synthesis. Nevertheless, most molecule generative models remain black-boxes, whose utilities are limited by a lack of interpretability and human participation in the generation process. In this work, we propose **Chem**ical **Spac**e **E**xplorer (ChemSpacE), a simple yet effective method for exploring the chemical space with pre-trained deep generative models. Our method enables users to interact with existing generative models and steer the molecule generation process. We demonstrate the efficacy of ChemSpacE on the molecule optimization task and the latent molecule manipulation task in single-property and multi-property settings. On the molecule optimization task, the performance of ChemSpacE is on par with previous black-box optimization methods yet is considerably faster and more sample efficient. Furthermore, the interface from ChemSpacE facilitates human-in-the-loop chemical space exploration and interactive molecule design. Code and demo are available at `https://github.com/yuanqidu/ChemSpacE`.

## 1 Introduction

Designing new molecules with desired properties is crucial for a wide range of tasks in drug discovery and materials science (Chen et al., 2018). Traditional pipelines for molecule design require exhaustive human effort and domain knowledge to explore the vast combinatorial chemical space, making them exceedingly difficult to scale. Recent studies exploit deep generative models to tackle this problem by encoding molecules into a meaningful latent space, from which random samples can be drawn and then decoded into new molecules. Such deep molecule generative models can facilitate the design and development of new drugs and materials (Lopez et al., 2020; Sanchez-Lengeling & Aspuru-Guzik, 2018).

Despite the promising results of deep generative models for molecule generation, much less effort has been made in understanding the internal representation of molecules and the underlying working mechanisms of these models, which are key to interpretable and interactive AI-empowered molecule design. Most existing models are based on deep neural networks or black-box optimization methods, which lack transparency and interpretability (Samek et al., 2019). Outside of the molecule generation domain, many attempts have been made to improve the interpretability of deep learning models from various aspects, *e.g.*, representation space (Zhou et al., 2016), model space (Guo et al., 2021), and latent space (Shen et al., 2020; Shen & Zhou, 2021). In molecule generation, interpretability can be studied from two perspectives: (1) the interpretation of the **learned latent space** where traversing the value of latent vectors could lead to smooth molecular property change, and (2) the interpretation of the **chemical space** where adjusting molecular properties could further provide insight to smooth and consistent structure change of molecules (i.e. structure-activity relationships).

Furthermore, it remains difficult to generate molecules with desired properties. Previous works tackle the problem with reinforcement learning-based, latent space optimization-based, and searching-based methods to achieve property control of the generated molecules (Shi et al., 2020; Jin et al., 2018a). Specifically, reinforcement learning-based algorithms (You et al., 2018a) equip the model with rewards designed to encourage the models to generate molecules with specific molecular properties. Latent space optimization-based algorithms take advantage of the learned latent space of molecule generative models and optimize the molecular properties via Bayesian Optimization (Liu et al., 2018). Searching-based algorithms, on the other hand, directly search the discrete, high-dimensional chemical space for molecules with optimal properties (Kwon et al., 2021). However, these works often have three major issues. (1) They require many expensive oracle calls to provide feedback (*i.e.*, property scores) on the intermediate molecules generated during the searching or optimization process (Huang et al., 2021). (2) They often only focus on the outcome of the process while ignoring its intermediate steps which can provide crucial insights into the rules that govern the process to chemists and pharmacologists. (3) They focus on local gradients and put less emphasis on exploring global directions in the chemical/latent space.

To tackle the above challenges, we propose a simple yet effective method to explore the chemical space for molecule generation by interpreting the latent space of pre-trained deep generative models. The motivation for our approach is based on the emergent properties of the latent space learned by molecule generative models (Gómez-Bombarelli et al., 2018; Zang & Wang, 2020): (1) molecules sharing similar structures/properties tend to cluster in the latent space, (2) interpolating two molecules in the latent space leads to smooth changes in molecular structures/properties. Thus, we develop *ChemSpace Explorer*, a model-agnostic method to manipulate molecules in the latent space of molecule generative models. It has broad applications ranging from molecule optimization to chemical space interpretation. Specifically, *ChemSpace Explorer* first identifies the *property separation hyperplane* which defines the binary boundary for some molecular property (*e.g.*, drug-like or drug-unlike) in the learned latent space of a generative model. Based on the identified property separation hyperplane, it then estimates the *latent directions* that govern molecular properties, navigating which can enable smooth change of molecular structures and properties without model re-training. This manipulation process improves the interpretability of deep generative models by navigating their latent spaces and enables *human-in-the-loop* exploration of the chemical space and molecule design. It allows users to manipulate the properties of generated molecules by leveraging the steerability and interpretability of molecule generative models. To the best of our knowledge, this work is the first attempt to achieve interactive molecule discovery by steering pre-trained molecule generative models.

Our experiments demonstrate that our method can effectively steer state-of-the-art molecule generative models for latent molecule manipulation with a small amount of training/inference time, data, and oracle calls. To quantitatively measure the performance of latent molecule manipulation, we design two new evaluation metrics, *strict success rate* and *relax success rate*, which explain the percentage of successful manipulation paths with smooth property-changing molecules. In addition, we compare ChemSpacE with a gradient-based optimization method that traverses the latent space of molecule generative models on the molecule optimization task. To facilitate interactive molecule design and discovery for practitioners, we further develop an interface for real-time interactive latent molecule manipulation and smooth molecular structure/property change. We summarize the main contributions as follows:

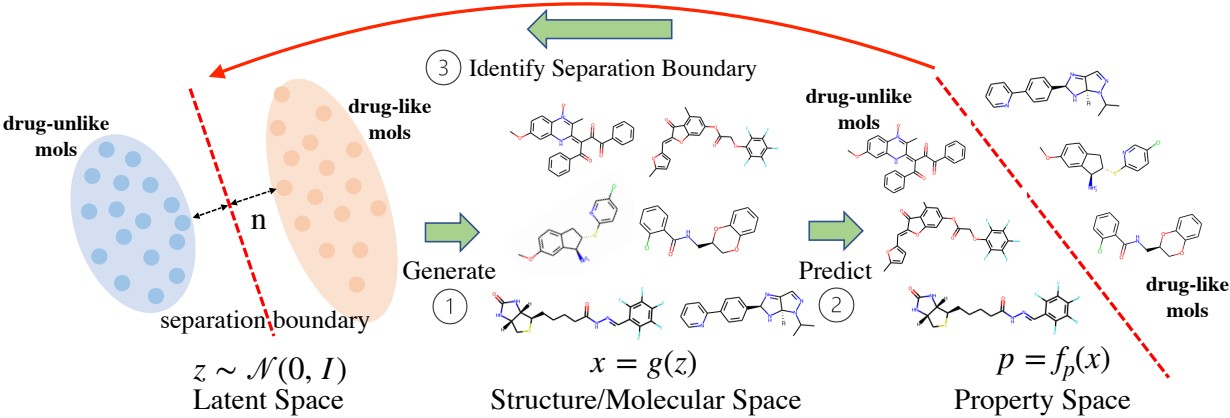

Figure 1: *ChemSpacE* framework: (1) a pre-trained molecule generative model generates many molecules from random vectors sampled from the latent space, (2) an off-the-shelf oracle function is used to predict molecular properties of the generated samples in the chemical space, (3) ChemSpacE identifies directions in the latent space which govern molecular properties via the property separation hyperplane.

- We explore a new task called *latent molecule manipulation*, which aims to steer the latent space of molecule generative models to manipulate the chemical properties of the output molecule and facilitate *human-in-the-loop* molecule design.

- We develop an efficient model-agnostic method named *ChemSpacE* for latent molecule manipulation, which can be incorporated in various pre-trained state-of-the-art molecule generative models without needing to re-train or modify them.

- We demonstrate the effectiveness and efficiency of our method in molecule optimization and achieving *human-in-the-loop* molecule design through comprehensive experiments. We further develop an interface to exhibit interactive molecule discovery and design.

## 2 Problem Formulation of Latent Molecule Manipulation

**Molecule Generative Models.** In molecule generation, a generative model $M$ encodes the molecular graph $X$ as a latent vector $Z \in \mathbb{R}^l$ where $l$ is the latent space dimension, and then decodes the latent vector back to the molecular graph. Variational auto-encoders (VAE) (Kingma & Welling, 2013) and flow-based models (Flow) (Rezende & Mohamed, 2015) are the two most commonly used models for molecule generation, which typically encode the data from the molecular space to a latent space of Gaussian distribution. The encoding and decoding process can be formulated as:

$$z = f(x), \qquad x' = g(z), \tag{1}$$

where $x$ and $x'$ are the ground-truth and reconstructed/sampled data respectively, and $z \in Z$ represents a latent vector in the latent space, $f(\cdot)$ and $g(\cdot)$ are the encoder and generator/decoder of the generative model. Note that we simplify the expression here to represent the general latent space that we seek to navigate in both VAEs and Flows. In practice, VAEs resort to a reparametrization trick such that $z = \mu + \sigma \odot \epsilon$, where $\epsilon \sim \mathcal{N}(0, I)$.

**Latent Molecule Manipulation Formulation.** To leverage the steerability and interpretability of molecule generative models, we explore a new task, *latent molecule manipulation*, which interprets and steers the latent space of generative models in order to manipulate the properties of the output molecules. To be specific, a deep generative model contains a generator $g : \mathcal{Z} \to \mathcal{X}$, where $\mathcal{Z} \in \mathbf{R}^l$ stands for the

$l$-dimensional latent space, which is commonly assumed to be a Gaussian distribution (Kingma & Welling, 2013; Rezende & Mohamed, 2015). There exist property functions $f_P$ which define the property space $\mathcal{P}$ via $P = f_P(X)$. The input to latent molecule manipulation is a list of $n$ molecules $X = \{x_1, x_2, \cdots, x_n\}$ and a list of $m$ molecular properties $P = \{p_1, p_2, \cdots, p_m\}$. We aim to manipulate one or more molecular properties $p$ of a given molecule in $k$ consecutive steps and output the manipulated molecules with properties $p' = \{p^{(1)}, p^{(2)}, \cdots, p^{(k)}\}$. By manipulating the given molecule, we can observe the alignment of $\mathcal{Z} \rightarrow \mathcal{X} \rightarrow \mathcal{P}$, where the relationship between $\mathcal{Z}$ and $\mathcal{X}$ explains the latent space of molecule generative models and the relationship between $\mathcal{X}$ and $\mathcal{P}$ reveals the correlations between molecular structures and properties. By traversing the latent space, we can generate molecules with continuous structure/property changes.

**Evaluation Criteria.** We need to evaluate the latent molecule manipulation task with respect to both smooth structure change and the smooth property change. Thus, we design two new evaluation metrics, *strict success rate (SSR)* and *relaxed success rate (RSR)*, that measure the quality of the identified latent direction in controlling the molecular property. For calculating the strict success rate, we consider a manipulation path to be successful only if we generate molecules with monotonically-changing properties and structures in consecutive $k$ steps of manipulation. The constraints are formulated as follows:

$$\phi_{SPC}(x, k, f) = 1[\forall\, i \in [k], s.t., f(x^{(i)}) - f(x^{(i+1)}) \leq 0], \tag{2}$$

$$\phi_{SSC}(x, k, \delta) = 1[\forall\, i \in [k], s.t., \delta(x^{(i+1)}, x^{(1)}) - \delta(x^{(i)}, x^{(1)}) \leq 0], \tag{3}$$

$$\phi_{DIV}(x, k) = 1[\exists\, i \in [k], s.t., x^{(i)} \neq x^{(1)}], \tag{4}$$

where $f$ is a property function which calculates certain molecular property, $\delta$ denotes structure similarity between molecules $x^{(i)}, x^{(i+1)}$ generated in two adjacent manipulation steps. $\phi_{SPC}$ defines the strict property constraint; $\phi_{SSC}$ defines the strict structure constraint; $\phi DIV$ defines the diversity constraint. The strict success rate is defined as:

$$SSR - L(P, X, k) = \frac{1}{|P| \times |X|} \sum_{p \in P, x \in X} 1[\phi_{SPC}(x_p, k, f_p) \wedge \phi_{SSC}(x_p, k) \wedge \phi_{DIV}(x_p, k)], \tag{5}$$

As monotonicity is rather strict, we propose a more relaxed definition of success rate, namely relaxed success rate, constructed via relaxed constraints, as follows:

$$\phi_{RPC}(x, k, f, \epsilon) = 1[\forall\, i \in [k], s.t., f(x^{(i)}) - f(x^{(i+1)}) \leq \epsilon], \tag{6}$$

$$\phi_{RSC}(x, k, \delta, \gamma) = 1[\forall\, i \in [k], s.t., \delta(x^{(i+1)}, x^{(1)}) - \delta(x^{(i)}, x^{(1)}) \leq \gamma], \tag{7}$$

$$\phi_{DIV}(x, k) = 1[\exists\, i \in [k], s.t., x^{(i)} \neq x^{(1)}], \tag{8}$$

where $\epsilon$ is a predefined tolerance threshold that weakens the monotonicity requirement. We also provide two implementations of relaxed success rate, which define different tolerance variables $\epsilon$, one with a local relaxed constraint (RSR-L) and the other with a global relaxed constraint (RSR-G). For the global constraint, we obtain $\epsilon$ by calculating the possible values (ranges) of the molecular properties in the training dataset, while for the local constraint, we obtain $\epsilon$ by calculating the possible values (ranges) of the molecular properties only in the specific manipulation paths. The formulation of RSR-L and RSR-G is as follows:

$$RSR - L(P, X, k, \epsilon_l, \gamma) = \frac{1}{|P| \times |X|} \sum_{p \in P, x \in X}$$
$$1[\phi_{RPC}(x_p, k, f_p, \epsilon_l) \wedge \phi_{RSC}(x_p, k, \gamma) \wedge \phi_{DIV}(x_p, k)], \tag{9}$$

$$RSR - G(P, X, k, \epsilon_g, \gamma) = \frac{1}{|P| \times |X|} \sum_{p \in P, x \in X}$$
$$1[\phi_{RPC}(x_p, k, f_p, \epsilon_g) \wedge \phi_{RSC}(x_p, k, \gamma) \wedge \phi_{DIV}(x_p, k)], \tag{10}$$

While it is more challenging for the model to pass RSR-L with a local constraint (smaller range) when evaluating a successful path, it has an added benefit in that it takes into account the ability of the model to manipulate one molecular property (*i.e.*, the larger the range, the higher the tolerance score, thus the better chance to achieve successful manipulation).

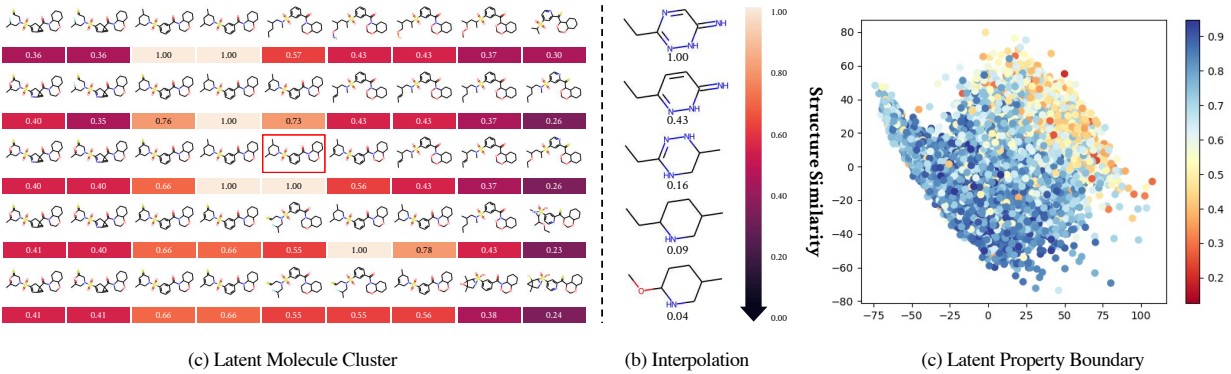

(c) Latent Molecule Cluster     (b) Interpolation     (c) Latent Property Boundary

Figure 2: (a) Molecule clusters in the latent space, where the number represents structure similarity (Bajusz et al., 2015). The red box denotes the base molecule and the x and y axes represent two random orthogonal directions to manipulate. (b) Linear interpolation of two (top and bottom) molecules. (c) The latent property boundary for QED is visualized for MoFlow trained on ZINC by reducing the dimension of the latent vectors by PCA.

## 3 ChemSpacE for Latent Molecule Manipulation

### 3.1 Latent Cluster Assumption

We examine the properties of the latent space learned by generative models and have the following observations: (1) molecules with similar structures tend to cluster together in the latent space, (2) interpolating two molecules $x_1$ and $x_2$, represented by latent vectors $z_1$ and $z_2$, can lead to a list of intermediate molecules whose structures/properties gradually change from $x_1$ to $x_2$. As molecular structures determine molecular properties (Seybold et al., 1987), the observations imply that molecules with similar values of a certain molecular property would cluster together, and interpolating between two molecules with different values of the molecular property could lead to gradual changes in their molecular structure. As shown in Fig. 1, there may exist two groups of molecules, drug-like and drug-unlike, where each group cluster together and linearly interpolating between two latent vectors with one molecule from each group could lead to a direction that crosses the property separation boundary. These observations also match the analysis from the prior work (Gómez-Bombarelli et al., 2018; Zang & Wang, 2020). To verify our assumption, we visualize the latent space of the pre-trained MoFlow model in Fig. 2. The left figure shows that molecules close together in the latent space are similar in structure, the middle figure shows that interpolating between two molecules in the latent space could lead to smooth structure changes, and the right figure shows that a latent boundary is present for the QED property in the pre-trained MoFlow model.

### 3.2 Identifying Latent Directions

**Latent Separation Boundary.** With the verifications shown above and the previous work of analyzing the latent space of generative models (Shen et al., 2020; Bau et al., 2017; Jahanian et al., 2019; Plumerault et al., 2020), we assume that there exists a separation boundary which separates groups of molecules for each molecular property (*e.g.*, drug-like and drug-unlike) and the normal vector of the separation boundary defines a latent direction which controls the degree of the property value (Fig. 1). When $z$ moves toward and crosses the boundary, the molecular properties change accordingly (*e.g.*, from drug-unlike to drug-like). A perfect separation boundary would have molecules with different properties well separated on different sides. From that, we can find a separation boundary for each molecular property with a unit normal vector $n \in \mathbf{R}^l$, such that the distance from any sample $z$ to the separation boundary as:

$$d(z, n) = n^T z. \tag{11}$$

**Latent Direction.** In the latent space, the molecular structure and property change smoothly towards the new property class when $z$ moves towards the separation boundary and vice versa, where we assume linear dependency between $z$ and $p$:

$$f_P(g(z)) = \alpha \cdot d(z, n), \tag{12}$$

where $f_P$ is an oracle function and $\alpha$ is a degree scalar that scales the changes along that corresponding direction. Extending the method to multiple molecular property manipulation, we have:

$$f_P(g(z)) = AN^T z, \tag{13}$$

where $A = Diag(a_1, \cdots, a_m)$ is the diagonal matrix with linear coefficients for each of the $m$ molecular properties and $N = [n_1, \cdots, n_m]$ represents normal vectors for the separation boundaries of $m$ molecular properties. We have the molecular properties $P$ following a multivariate normal distribution via:

$$\mu_P = \mathbf{E}(AN^T z) = AN^T \mathbf{E}(z) = \mathbf{0}, \tag{14}$$
$$\Sigma_P = \mathbf{E}(AN^T zz^T NA^T) = AN^T \mathbf{E}(zz^T)NA^T = AN^T NA^T. \tag{15}$$

We have all disentangled molecular properties in $P$ if and only if $\Sigma_P$ is a diagonal matrix and all directions in $N$ are orthogonal with each other. Nevertheless, not all molecular properties are purely disentangled with each other. In that case, molecular properties can correlate with each other and $n_i^T n_j$ is used to denote the entanglement between the $i$-th and $j$-th molecular properties in $P$.

### 3.3 Latent Molecule Manipulation

After we find the separation boundary and identify the latent direction, to manipulate the generated molecules towards desired properties, we first move from the latent vector $z$ along the direction $n$ with a degree scalar $\alpha$, giving the new latent vector

$$z' = z + \alpha n. \tag{16}$$

The expected property value of the new manipulated molecule is (with $k$ as a scale factor):

$$f_P(g(z + \alpha n)) = f_P(g(z)) + k\alpha. \tag{17}$$

For single-property manipulation, we can simply take the identified direction, but when multiple properties correlate with each other, we need to determine whether the two directions are entangled or disentangled. We can then take the disentangled and positively correlated attributes of the directions as the new direction:

$$n = n_1 + (1_{[n_1 \odot n_2 \geq 0]}) \odot n_2. \tag{18}$$

## 4 Experiments

### 4.1 Setup

**Datasets.** We use three molecule datasets, QM9 (Ramakrishnan et al., 2014), ZINC250K (Irwin & Shoichet, 2005), and ChEMBL (Mendez et al., 2019). QM9 contains 134k small organic molecules with up to 9 heavy atoms (C, O, N, F). ZINC250K (Gómez-Bombarelli et al., 2018) is a sampled set of 250K molecules from ZINC, a free database of commercially-available compounds for drug discovery with an average of $\sim 23$ heavy atoms. ChEMBL is a manually curated database of bioactive molecules with drug-like properties and contains $\sim 1.8$ million molecules.

**Baselines.** We include two baseline methods of identifying the latent direction that governs the molecular property and one gradient-based method, which optimizes the molecular property of the generated molecules via gradient ascent/descent for comparison. **Random manipulation** randomly samples latent directions for molecular properties. **Largest range manipulation** draws latent vectors from the training set and defines the directions via calculating the direction between one molecule with the largest property score

Table 1: Quantitative Evaluation of latent molecule manipulation over a variety of molecular properties (numbers reported are *strict success rate* in %. The best performances are bold.)

| Dataset | | Model | Avg. | QED | pLogP | SA | DRD2 | JNK3 | GSK3B | MolWt |
|---------|---|-------|------|-----|-------|-----|------|------|-------|-------|
| QM9 | MoFlow | Random | 1.65 | 1.50 | 0.00 | 0.50 | 0.00 | 0.00 | 0.00 | 0.50 |
| | | Largest | 3.43 | 1.50 | 1.00 | 0.50 | 0.00 | 1.50 | 0.00 | 0.50 |
| | | Gradient-based | N/A | 3.50 | 6.00 | 6.50 | 2.00 | 8.00 | 8.50 | 7.50 |
| | | ChemSpacE | **37.52** | **12.50** | **9.00** | **10.00** | **11.00** | **45.50** | **16.50** | **10.50** |
| | HierVAE | Random | 29.29 | 1.00 | 1.50 | 0.50 | 0.50 | 1.00 | 1.00 | 0.50 |
| | | Largest | 30.69 | 0.50 | 0.00 | 0.00 | 0.50 | 2.00 | 0.00 | 0.50 |
| | | ChemSpacE | **66.23** | **27.00** | **32.00** | **35.00** | **41.50** | **51.50** | **30.00** | **39.50** |
| ZINC | MoFlow | Random | 4.25 | 1.50 | 1.50 | 2.50 | 3.00 | 3.50 | 1.50 | 2.00 |
| | | Largest | 5.61 | 1.50 | 6.50 | 2.00 | 6.00 | 2.50 | 4.00 | 1.50 |
| | | Gradient-based | N/A | 1.50 | 9.50 | 0.50 | 2.00 | 15.50 | 23.00 | 0.00 |
| | | ChemSpacE | **58.08** | **52.00** | **53.50** | **51.50** | **55.00** | **56.50** | **55.50** | **53.50** |
| ChEMBL | HierVAE | Random | 25.59 | 0.00 | 0.00 | 0.00 | 0.00 | 0.00 | 0.00 | 0.00 |
| | | Largest | 22.98 | 0.00 | 0.00 | 0.00 | 0.00 | 0.00 | 0.00 | 0.00 |
| | | ChemSpacE | **47.70** | **0.50** | **3.00** | **3.00** | **6.00** | **7.50** | **5.50** | **4.50** |

and another molecule with the smallest property score for each molecular property. The **gradient-based method** optimizes the molecular property of the generated molecules by searching for a latent vector with the optimized molecular property via gradient ascent/descent. More specifically, it requires pre-training a property predictor on the latent space that first initializes a latent vector and then optimizes the latent vector to maximize/minimize the output of the predicted property value.

**Molecular Properties.** QED is a quantitative estimate of drug-likeness. PLogP refers to the partition coefficient logarithm of octanol-water which measures the lipophilicity and water solubility. SA denotes the synthesis accessibility score. MolWt denotes the molecular weight. DRD2, JNK3 and GSK3B are three binding affinity scores.

**Implementation Details.** We take publicly available pre-trained models from their respective GitHub repositories for HierVAE (Jin et al., 2020) and MoFlow (Zang & Wang, 2020). All the molecular properties are calculated by RDKit (Landrum et al., 2013) and TDC (Huang et al., 2021). We utilize the implementation of linear models (linear SVM) from Scikit-learn (Pedregosa et al., 2011). More details are available in Appendix A.

**Interactive Demo.** An interactive demo for latent molecule manipulation is provided at `https://drive.google.com/drive/folders/1N036p_5OfvGZybgPJ3Vw1ONXHVepimSR?usp=sharing` and one example is shown in Fig. 4 (right).

## 4.2 Quantitative Evaluation of latent molecule manipulation

In Tables 1 and 2, we report the quantitative evaluation results for both single property and multi-property latent molecule manipulation. Table 1 shows the strict success rate and relaxed success rate-L/G while Table 2 reports the training and inference times, data requirements, and number of oracle calls from evaluations on 212 molecular properties over 200 randomly generated molecules. From the tables, we obtain the following insights:

(1) Our proposed method, ChemSpacE, as the first attempt for latent molecule manipulation, achieves excellent performance in both single and multi-property manipulation of molecules with two state-of-the-art molecule generative models (VAE-based and Flow-based). For some important molecular properties (*e.g.,*

Table 2: Efficiency in terms of training/inference time, data, and number of oracle calls of ChemSpacE compared to the gradient-based method.

| Model | Dataset | Training(s) | Inference/Path(s) | # Data | # Oracle calls |
|---|---|---|---|---|---|
| Gradient-based | QM9 | 137.03 | 0.02 | 120k | 120k |
| | ZINC | 1027.26 | 0.04 | 200k | 200k |
| ChemSpacE | QM9 | 0.05 | 0 | 300 | 300 |
| | ZINC | 0.95 | 0 | 400 | 400 |
| Speedup | QM9 | 2740× | 0.02 ↑ | 400× | 400× |
| | ZINC | 1080× | 0.04 ↑ | 500× | 500× |

Table 3: Single property constrained molecule optimization for Penalized-logP on the ZINC dataset with four comparison methods ($\delta$ is the threshold for similarity between the optimized and base molecules).

| | $\delta$ | MoFlow | | | ChemSpacE | | |
|---|---|---|---|---|---|---|---|
| | | **Improvement** | **Similarity** | **Success** | **Improvement** | **Similarity** | **Success** |
| pLogP | **0.0** | $8.61 \pm 5.44$ | $0.30 \pm 0.20$ | 98.88% | $9.94 \pm 6.09$ | $0.18 \pm 0.14$ | 100% |
| | **0.2** | $7.06 \pm 5.04$ | $0.43 \pm 0.20$ | 96.75% | $7.17 \pm 5.59$ | $0.42 \pm 0.21$ | 96.00% |
| | **0.4** | $4.71 \pm 4.55$ | $0.61 \pm 0.18$ | 85.75% | $4.16 \pm 4.43$ | $0.65 \pm 0.20$ | 84.38% |
| | **0.6** | $2.10 \pm 2.86$ | $0.79 \pm 0.14$ | 58.25% | $1.76 \pm 2.40$ | $0.81 \pm 0.15$ | 59.63% |
| DRD2 | **0.0** | $9.99 \times 10^{-3} \pm 2.82 \times 10^{-2}$ | $0.29 \pm 0.17$ | 100% | $2.12 \times 10^{-2} \pm 1.84 \times 10^{-2}$ | $0.05 \pm 0.06$ | 100% |
| | **0.2** | $7.66 \times 10^{-3} \pm 2.66 \times 10^{-2}$ | $0.36 \pm 0.13$ | 100% | $5.49 \times 10^{-3} \pm 1.46 \times 10^{-2}$ | $0.34 \pm 0.14$ | 99.13% |
| | **0.4** | $1.24 \times 10^{-3} \pm 2.36 \times 10^{-3}$ | $0.52 \pm 0.12$ | 98.60% | $1.04 \times 10^{-3} \pm 1.83 \times 10^{-3}$ | $0.57 \pm 0.16$ | 95.75% |
| | **0.6** | $1.67 \times 10^{-4} \pm 4.10 \times 10^{-4}$ | $0.78 \pm 0.14$ | 85.20% | $1.79 \times 10^{-4} \pm 4.15 \times 10^{-4}$ | $0.80 \pm 0.15$ | 83.00% |
| JNK3 | **0.0** | $2.75 \times 10^{-2} \pm 2.22 \times 10^{-2}$ | $0.39 \pm 0.21$ | 99.40% | $4.79 \times 10^{-2} \pm 2.15 \times 10^{-2}$ | $0.19 \pm 0.15$ | 100% |
| | **0.2** | $2.34 \times 10^{-2} \pm 2.04 \times 10^{-2}$ | $0.44 \pm 0.19$ | 98.80% | $3.24 \times 10^{-2} \pm 2.21 \times 10^{-2}$ | $0.39 \pm 0.17$ | 99.38% |
| | **0.4** | $1.33 \times 10^{-2} \pm 1.54 \times 10^{-2}$ | $0.60 \pm 0.16$ | 95.60% | $1.94 \times 10^{-2} \pm 1.88 \times 10^{-2}$ | $0.58 \pm 0.15$ | 97.13% |
| | **0.6** | $6.27 \times 10^{-3} \pm 1.04 \times 10^{-2}$ | $0.79 \pm 0.16$ | 77.80% | $9.27 \times 10^{-3} \pm 1.38 \times 10^{-2}$ | $0.76 \pm 0.14$ | 85.00% |
| GSK3$\beta$ | **0.0** | $5.09 \times 10^{-2} \pm 4.35 \times 10^{-2}$ | $0.40 \pm 0.22$ | 98.60% | $1.21 \times 10^{-1} \pm 4.82 \times 10^{-2}$ | $0.15 \pm 0.12$ | 100% |
| | **0.2** | $4.21 \times 10^{-2} \pm 3.72 \times 10^{-2}$ | $0.47 \pm 0.19$ | 97.40% | $7.66 \times 10^{-2} \pm 5.01 \times 10^{-2}$ | $0.35 \pm 0.15$ | 99.50% |
| | **0.4** | $2.87 \times 10^{-2} \pm 3.02 \times 10^{-2}$ | $0.58 \pm 0.15$ | 95.20% | $3.87 \times 10^{-2} \pm 3.64 \times 10^{-2}$ | $0.57 \pm 0.16$ | 97.63% |
| | **0.6** | $1.34 \times 10^{-2} \pm 2.34 \times 10^{-2}$ | $0.76 \pm 0.14$ | 85.60% | $1.54 \times 10^{-2} \pm 2.28 \times 10^{-2}$ | $0.78 \pm 0.15$ | 86.63% |

QED), we (with MoFlow) achieve a 52% manipulation strict success rate on the ZINC dataset, outperforming the baseline methods 6× on average.

(2) The baseline (random manipulation) method sometimes "finds" directions that control molecular properties. As shown in Fig. 2, the molecules are well-clustered in the latent space with respect to structures that determine molecular properties (Seybold et al., 1987). However, the largest range manipulation method performs worse possibly due to its strong assumption in determining the direction via the molecules with extreme property values (largest property and smallest property values) in the dataset.

(3) The ChemSpacE method outperforms the popular gradient-based method in generating smooth manipulation paths, time, and data efficiency. As shown in Table 2, ChemSpacE speeds up the training time by at least 1000×, and reduces data requirements and the number of oracle calls necessary by at least 400×.

More results can be found in Appendix Tables 5 and 6.

### 4.3 Quantitative Evaluation of Molecule Optimization

We further compare our methods under the common molecule optimization setting through two tasks: *single property constrained optimization* and *multi-property constrained optimization*. Beginning with a set of candidate molecules, we aim to optimize their molecular properties while maintaining the structural similarity[1] of the optimized molecules to the base molecules. This setting is relevant to many drug discovery tasks where one needs to optimize the properties of a given molecule while keeping the structure similar.

**Single Property Constrained Optimization.** We evaluate our method against four previous works (Jin et al., 2018a; You et al., 2018a; Zang & Wang, 2020; Eckmann et al., 2022) with the exact same set of molecules

---
[1]In practice, we use the Tanimoto similarity of the Morgan fingerprint Rogers & Hahn (2010).

on the penalized logP property and test four different similarity constraint thresholds. We report the property improvement and similarity compared to the base molecule as well as the percentage of successfully optimized molecules within the threshold in Table 3. In addition, we evaluate on three more real-world properties, activities at the targets DRD2, JNK3, and GSK3$\beta$, under the same constrained optimization setting. In our reported results, ChemSpacE is manipulating over the latent space learned by MoFlow, as MoFlow leverages a gradient-based method that traces the local gradient leading to property improvements in every step while we take on a more efficient way by learning the global improvement direction and following it for all steps. We perform surprisingly well and even better than the gradient-based method used in MoFlow, thus empirically supporting our assumption about latent space exploration.

plogp: -5.200 QED: 0.360    plogp: -1.671 QED: 0.645

plogp: -6.366
QED: 0.449    plogp: -2.726
QED: 0.809

Figure 3: Illustrations of multi-property constrained optimization. The Tanimoto similarity between base and optimized molecules is 0.709 (top row) and 0.647 (bottom row) respectively.

**Multi-Property Constrained Optimization**. As no previous work reports on multoi-property constrained optimization, we propose to optimize QED and penalized logP as a multi-property constrained optimization task. We also propose two simple baselines: (1) we add up the two properties (QED and penalized logP) to be optimized as a new objective and run single-property constrained optimization on it, (2) we take into account the two gradient directions of the properties and in each step of gradient ascent, we move in both directions. As shown in Appendix (Table 8), we demonstrate the capability of ChemSpacE for efficient multi-objectiven constrained optimization. Our method improves both QED and penalized logP more than the two gradient-based methods. We showcase two examples in Fig. 3 that demonstrate ChemSpacE's ability to optimize molecules and achieve desired property improvements while maintaining high structure perseverance.

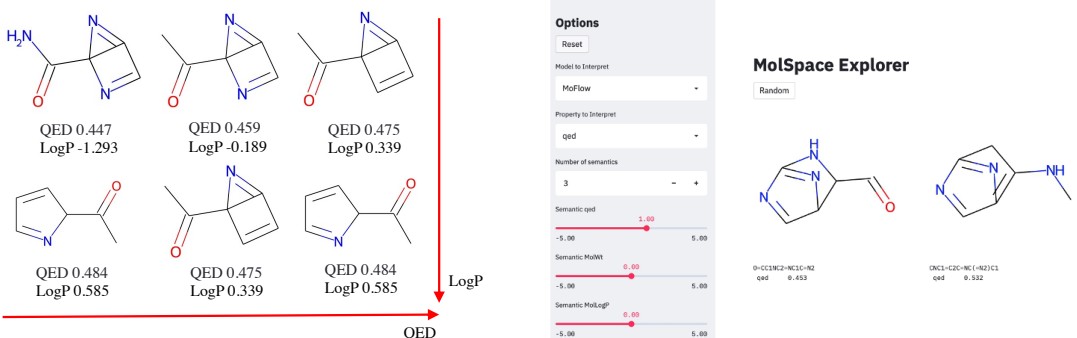

Figure 4: Manipulating QED and LogP properties of sampled molecules simultaneously with the MoFlow model trained on the QM9 dataset (the repeated molecules are removed for better visualization) (left). A Real-time Interactive System Interface. Please refer to the Appendix D video for a demo of interactive molecule discovery (right).

### 4.4 Qualitative Evaluation of Latent Molecule Manipulation and Interpretation

In Fig. 5, we visualize the property distributions of QED, MolWt and LogP along a 7-step manipulation path. For each step, we draw a property distribution. The candidate molecules are at place 0 and we attempt to manipulate the molecular property to the left (lower) and the right (higher). From the figure, we can clearly observe that the property distribution shifts to the left and right accordingly when we manipulate the molecule to the left and right. For example, when we manipulate the molecules three steps to the left, the range of QED shifts from $[0, 0.7]$ to $[0, 0.5]$; when the molecules are manipulated three steps to the right, there are far more molecules that have QED $> 0.5$ than the base distribution. Similar trends can also be seen for MolWt and LogP.

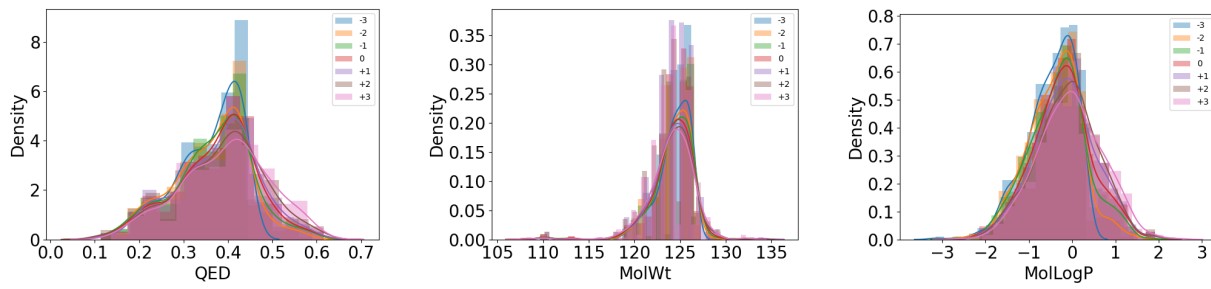

Figure 5: Visualization of molecular property distribution shifts while manipulating molecules with MoFlow on the QM9 dataset (0 denotes the randomly sampled base molecule and $+x$ and $-x$ denote manipulation directions and steps).

**Single Property Manipulation.** To qualitatively evaluate the performance of our method for latent molecule manipulation, we randomly select the successful manipulation paths from all three generative models in Fig. 6. The figures show that our method successfully learns interpretable and steerable directions. For example, for HierVAE in Fig. 6, we can find that gradually increasing LogP of a molecule may lead to the removal of the heavy atoms $O$ and $N$ from the structure. With respect to QED, the molecule drops double bonds, as well as heavy $N$ and $O$ atoms, when increasing QED for the HierVAE model. A similar trend can be observed in the MoFlow model that increasing QED drops double bonds and $O$ atoms on the left of Fig. 6.

**Multi-Property Manipulation.** When it comes to multi-property manipulation, the goal is to control multiple molecular properties of a given molecule at the same time. In Fig. 4 (left), we show how our method manipulates multiple molecular properties. For simplicity, we remove the duplicate molecules and only leave the distinct molecules during the manipulation. From the figure, we can observe some correlations between LogP and QED since when we increase QED, LogP also increases accordingly.

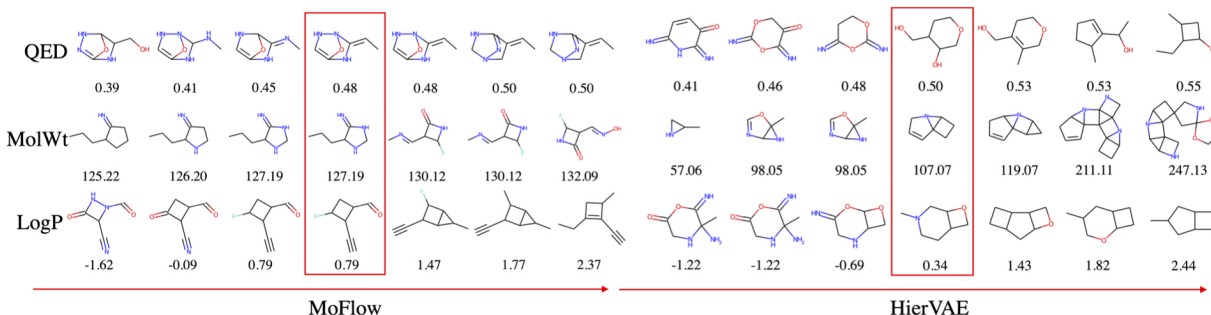

Figure 6: Manipulating QED, MolWt and LogP properties of sampled molecules. The backbone model is MoFlow and HierVAE trained on QM9 dataset.

## 5 Related Work

**Molecule Generation.** Recent studies have explored a variety of deep generative models for molecule generation (Du et al., 2022), including variational autoencoders (VAEs) (Jin et al., 2018a), generative adversarial networks (GANs) (De Cao & Kipf, 2018), reinforcement learning models (Olivecrona et al., 2017; Zhou et al., 2019; Yang et al., 2021), *etc* (Yang et al., 2020; Xie et al., 2021), normalizing flows (Madhawa et al., 2019; Shi et al., 2020; Luo et al., 2021), and energy-based models (EBMs) (Liu et al., 2021). For instance, JT-VAE (Jin et al., 2018a) proposes a VAE-based architecture to encode both atomic graphs and structural graphs for efficient molecule generation. MolGAN (De Cao & Kipf, 2018) exploits GANs for molecule generation, where discriminators are used to encourage the model to generate realistic and chemically-valid molecules. MRNN (Popova et al., 2019) extends the idea of GraphRNN (You et al., 2018b) to formulate molecule generation as an auto-regressive process. GCPN (You et al., 2018a) formulates the molecule generation process as a reinforcement learning problem where it connects atoms in a step-by-step fashion to obtain a molecule and uses reward for steerable generation. GraphNVP (Madhawa et al., 2019) introduces normalizing flows for molecule generation, where the generation process is invertible. Later works improve nomralizing flow-based molecule generative models by introducing auto-regressive generation (Shi et al., 2020), valency correction (Zang & Wang, 2020), and discrete latent representation (Luo et al., 2021). GraphEBM (Liu et al., 2021) introduces an energy-based model for molecule generation. More recently, diffusion models have been applied to molecule generation (Niu et al., 2020; Jo et al., 2022; Vignac et al., 2022).

**Controllable Molecule Generation.** One practical application of molecule generation is to generate new molecular samples with specific properties. Early work (Segler et al., 2018) involved biasing the distribution of the data and training generative models with known desired properties to generate molecules with those properties. In contrast, recent works have focused on leveraging latent space gradient-based (Jin et al., 2018a; You et al., 2018a; Hoffman et al., 2020; Winter et al., 2019), reinforcement learning-based (Shi et al., 2020; Zang & Wang, 2020; Blaschke et al., 2020), and searching-based (Brown et al., 2019; Yang et al., 2020; Kwon et al., 2021) approaches to generate molecules with desired properties. Latent space gradient-based methods are flexible and can work directly on both molecules (Fu et al., 2022) and the learned latent vectors (Gómez-Bombarelli et al., 2018; Jin et al., 2018b; Winter et al., 2019; Griffiths & Hernández-Lobato, 2020; Notin et al., 2021). Reinforcement learning-based methods usually formulate controllable generation as a sequential decision-making problem and require a score-function to reward the agent. Searching-based approaches (Brown et al., 2019; Yang et al., 2020; Kwon et al., 2021; Renz et al., 2019; Fu et al., 2020; Xie et al., 2021; Maziarz et al., 2021) can search over the chemical space for molecules with desired properties by defining a set of discrete actions. Additionally, some works (Chenthamarakshan et al., 2020; Das et al., 2021) leverage the learned latent space and achieve controllable generation by accepting/rejecting sampled molecules based on a molecular property predictor.

## 6 Conclusion, Limitations and Future Work

In this work, we propose a simple yet effective method called ChemSpacE to generate molecules with desired properties by leveraging the steerability and interpretability of pre-trained generative models. Furthermore, our interface demonstrates promising applications of interactive molecule design and discovery. Nevertheless, we acknowledge two limitations of this work, (1) it cannot yet study the activity cliff phenomenon, (2) it lacks theoretical analyses explaining why the latent space of deep generative models can be learned with property boundaries. Studying the activity cliff, a phenomenon in which structurally similar molecules may have very different potencies against the same target (Stumpfe et al., 2014) is a very challenging task that requires specific benchmark datasets, and reliable oracle functions for molecule generation, making it beyond the scope of this study. We expect that an enhanced understanding of the chemical space will lead to promising directions of study in understanding such challenging phenomena. Furthermore, although it has been widely observed that semantic directions can be found in the latent space of generative models and exploited for data editing, we believe that the theoretical underpinnings of these observations are not well understood and merit further study. Finally, we hope ChemSpacE opens up a new research avenue to study the interpretability and steerability of molecule generative models for achieveing interactive design.

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

# Appendix for
# "ChemSpacE: Interpretable and Interactive Chemical Space Exploration"

## A    Experiment Protocols

**Pre-trained Models.** We apply ChemSpacE, as well as baselines, on two state-of-the-art molecule generative models with publicly available pre-trained models. HierVAE (Jin et al., 2020) embeds molecular structure motifs into a hierarchical VAE-based generative model; MoFlow (Zang & Wang, 2020) designs a normalizing flow-based model which learns an invertible mapping between input molecules and latent vectors. **Molecular Properties.** We study molecular properties identified in the chemistry community through open-source cheminformatics software, RDKit (Landrum et al., 2013) and protein binding affinity, synthesis accessibility oracles in TDC (Huang et al., 2021). In total, we analyze 212 molecular properties from multiple dimensions, including distributions, inter-correlations, etc. Details can be found in Appendix G. Due to the page limit, we mainly report results for 7 molecular properties, including 4 very common yet important ones, drug-likeness (QED), molecular weight (MolWt), partition coefficient (LogP), synthesis accessibility (SA), and 3 binding affinity scores. For continuous molecular properties, we take the molecules with largest and smallest properties for training the linear models.

Quantitatively, we evaluate the ability of the model to manipulate the given molecular property of molecules with the proposed **strict success rate** and **relaxed success rate-L/G** metrics (see Sec. 2). We evaluate the model's efficiency by comparing the training process of the linear models with a neural network-based predictor for a commonly used optimization-based method in terms of training/inference time, data, and number of oracle calls. Qualitatively, we visualize latent molecule manipulation including property distribution shift during manipulation, single and multiple property manipulations.

**Hyperparameters.** ChemSpacE does not entail many hyperparameters, the only important one is the manipulation range which is critical to the exploration degree of the latent space. For latent molecule manipulation experiments, as we would like a gradual change over the molecular structure and property, we set the range as $[-1, 1]$. While for molecule optimization task, it requires more aggressive exploration strategies to reach the expected latent area which poses optimal property values. We utilize $[-100, 100]$ and $[-30, 30]$ for single property optimization and multi-property optimization experiments respectively. We report the results for single property optimization with ranges from $[1, 5, 10, 15, 20, 30, 50, 100]$ in Table 4.

## B    Extended Latent Molecule Manipulation Experiments

### B.1    Molecule Generation Evaluation

We evaluate the **Validity**, **Novelty** and **Uniqueness** of the generated molecules as proposed in Kusner et al. (2017) in Table 9. We can observe that ChemSpacE not only improves the success rate from the baseline methods, but also in general improves the novelty and uniqueness during manipulation.

### B.2    Multi-property Latent Molecule Manipulation Evaluation

We evaluate multi-property (penalized logp, QED) latent molecule manipulation over 200 randomly sampled molecules on ZINC dataset in Table 5.

## C    Extended Molecule Optimization Experiments

We report more experiments about single property and multi-property optimization in this section. In Table 4, pushing further across the property separation boundary increases the improvement for molecule optimization but lowers the similarity scores.

Table 4: Single property constrained molecule optimization for Penalized-logP on ZINC dataset with different manipulation ranges of ChemSpacE ($\delta$ is the threshold for similarity between the optimized and base molecules).

| | ChemSpacE-1 | | | ChemSpacE-5 | | |
|---|---|---|---|---|---|---|
| $\delta$ | **Improvement** | **Similarity** | **Success** | **Improvement** | **Similarity** | **Success** |
| **0.0** | $2.61 \pm 2.55$ | $0.71 \pm 0.23$ | 83.25% | $3.33 \pm 3.74$ | $0.67 \pm 0.26$ | 84.25% |
| **0.2** | $2.56 \pm 2.51$ | $0.72 \pm 0.22$ | 97.1% | $3.17 \pm 3.60$ | $0.69 \pm 0.23$ | 84.13% |
| **0.4** | $2.26 \pm 2.28$ | $0.75 \pm 0.20$ | 77.25% | $2.62 \pm 3.08$ | $0.73 \pm 0.20$ | 78.13% |
| **0.6** | $1.34 \pm 1.54$ | $0.84 \pm 0.14$ | 57.0% | $1.43 \pm 1.54$ | $0.84 \pm 0.14$ | 57.38% |

| | ChemSpacE-10 | | | ChemSpacE-15 | | |
|---|---|---|---|---|---|---|
| $\delta$ | **Improvement** | **Similarity** | **Success** | **Improvement** | **Similarity** | **Success** |
| **0.0** | $4.97 \pm 4.86$ | $0.57 \pm 0.27$ | 90.75% | $5.92 \pm 5.11$ | $0.51 \pm 0.26$ | 93.75% |
| **0.2** | $4.70 \pm 4.71$ | $0.60 \pm 0.24$ | 90.13% | $5.62 \pm 5.05$ | $0.55 \pm 0.23$ | 93.25% |
| **0.4** | $3.43 \pm 3.96$ | $0.69 \pm 0.20$ | 82.38% | $3.96 \pm 4.28$ | $0.73 \pm 0.20$ | 84.25% |
| **0.6** | $1.67 \pm 2.32$ | $0.82 \pm 0.15$ | 59.00% | $1.73 \pm 2.35$ | $0.81 \pm 0.15$ | 59.63% |

| | ChemSpacE-20 | | | ChemSpacE-30 | | |
|---|---|---|---|---|---|---|
| $\delta$ | **Improvement** | **Similarity** | **Success** | **Improvement** | **Similarity** | **Success** |
| **0.0** | $6.62 \pm 5.57$ | $0.46 \pm 0.25$ | 94.40% | $7.77 \pm 6.34$ | $0.39 \pm 0.24$ | 96.38% |
| **0.2** | $6.11 \pm 5.14$ | $0.51 \pm 0.22$ | 93.75% | $6.50 \pm 5.40$ | $0.48 \pm 0.22$ | 94.50% |
| **0.4** | $4.22 \pm 4.50$ | $0.65 \pm 0.19$ | 85.13% | $4.47 \pm 4.73$ | $0.64 \pm 0.19$ | 85.88% |
| **0.6** | $1.79 \pm 2.36$ | $0.81 \pm 0.15$ | 59.88% | $1.78 \pm 2.37$ | $0.81 \pm 0.15$ | 60.25% |

| | ChemSpacE-50 | | | ChemSpacE-100 | | |
|---|---|---|---|---|---|---|
| $\delta$ | **Improvement** | **Similarity** | **Success** | **Improvement** | **Similarity** | **Success** |
| **0.0** | $8.80 \pm 6.35$ | $0.30 \pm 0.21$ | 98.38% | $9.94 \pm 6.09$ | $0.18 \pm 0.14$ | 100% |
| **0.2** | $6.99 \pm 5.53$ | $0.44 \pm 0.21$ | 95.00% | $7.17 \pm 5.59$ | $0.42 \pm 0.21$ | 96.00% |
| **0.4** | $4.45 \pm 4.65$ | $0.63 \pm 0.19$ | 85.38% | $4.16 \pm 4.43$ | $0.65 \pm 0.20$ | 84.38% |
| **0.6** | $1.87 \pm 2.56$ | $0.80 \pm 0.15$ | 60.13% | $1.76 \pm 2.40$ | $0.81 \pm 0.15$ | 59.63% |

Table 5: Quantitative Evaluation of latent molecule manipulation for Multiple Properties. (-R denotes model with random manipulation, MoFlow-1 and MoFlow-2 denote two variants of gradient-based baseline methods, RSR(L) denotes *relaxed success rate-L*, RSR(G) denotes *relaxed success rate-G*).

| Metric | MoFlow-1 | MoFlow-2 | **ChemSpacE** |
|---|---|---|---|
| SSR-both | 28.00 | 27.00 | **62.00** |
| RSR(L)-both | 29.50 | 28.00 | **63.00** |
| RSR(G)-both | 41.00 | 38.50 | **76.00** |

Table 6: Quantitative Evaluation of latent molecule manipulation over a variety of molecular properties (numbers reported are *relaxed success rate-L / relaxed success rate-G* in %. The best performances are bold.)

| Dataset | | Model | Avg. | QED | LogP | SA | DRD2 | JNK3 | GSK3B | MolWt |
|---|---|---|---|---|---|---|---|---|---|---|
| QM9 | MoFlow | Random | 27.21 / 32.31 | 1.50 / 2.00 | 0.00 / 3.00 | 1.00 / 3.00 | 0.00 / 46.00 | 4.00 / 4.00 | 0.00 / 15.50 | 1.50 / 55.00 |
| | | Largest | 29.28 / 35.20 | 3.00 / 8.00 | 1.00 / 7.00 | 1.00 / 2.00 | 0.50 / 42.50 | 6.00 / 6.00 | 0.50 / 7.50 | 1.00 / 58.00 |
| | | Gradient-based | N/A | 4.50/6.50 | 6.50/8.50 | 8.50/13.00 | 3.00/15.0 | 10.50/10.50 | 10.50/17.50 | 8.50/22.00 |
| | | ChemSpacE | **53.97 / 61.56** | **16.00 / 28.00** | **13.50 / 28.00** | **17.50 / 39.50** | **17.50 / 72.50** | **58.50 / 58.50** | **21.50 / 49.00** | **15.00 / 72.00** |
| | HierVAE | Random | 2.62 / 26.06 | 1.00 / 1.00 | 1.50 / 1.50 | 0.50 / 0.50 | 0.50 / 1.50 | 1.00 / 5.50 | 1.00 / 3.00 | 0.50 / 2.50 |
| | | Largest | 3.25 / 27.33 | 0.50 / 1.00 | 0.00 / 1.50 | 0.00 / 5.50 | 0.50 / 4.00 | 2.00 / 8.50 | 0.00 / 2.50 | 0.50 / 1.50 |
| | | ChemSpacE | **46.72 / 61.49** | **27.00 / 35.00** | **32.00 / 44.00** | **35.00 / 42.00** | **41.50 / 48.50** | **51.50 / 60.00** | **30.00 / 33.50** | **39.50 / 45.50** |
| ZINC | MoFlow | Random | 35.85 / 41.70 | 3.50 / 6.00 | 2.50 / 7.50 | 3.50 / 6.50 | 5.50 / 79.00 | 4.00 / 56.50 | 1.50 / 27.50 | 4.50 / 12.50 |
| | | Largest | 37.46 / 43.12 | 3.00 / 4.50 | 9.00 / 15.50 | 2.00 / 6.00 | 8.00 / 81.50 | 4.00 / 67.50 | 4.00 / 33.00 | 3.00 / 14.50 |
| | | Gradient-based | N/A | 1.50/2.00 | 10.50/15.50 | 1.00/2.50 | 2.50/5.50 | 18.00/21.50 | 23.50/28.50 | 0.50/1.50 |
| | | ChemSpacE | **60.54 / 63.23** | **53.50 / 57.00** | **57.00 / 73.50** | **54.00 / 61.50** | **55.50 / 65.50** | **57.50 / 63.50** | **56.00 / 68.00** | **56.00 / 71.00** |
| ChEMBL | HierVAE | Random | 0.24 / 18.20 | 0.00 / 0.00 | 0.00 / 0.50 | 0.00 / 0.50 | 0.00 / 2.00 | 0.00 / 0.00 | 0.00 / 1.00 | 0.00 / 0.00 |
| | | Largest | 0.25 / 17.88 | 0.00 / 0.00 | 0.00 / 2.50 | 0.00 / 0.00 | 0.00 / 0.50 | 0.00 / 1.00 | 0.00 / 0.00 | 0.00 / 2.00 |
| | | ChemSpacE | **13.76 / 36.26** | **0.50 / 2.50** | **3.00 / 3.50** | **3.00 / 5.00** | **6.00 / 11.00** | **7.50 / 15.00** | **5.50 / 9.00** | **4.50 / 9.00** |

# D  ChemSpacE Demo

As shown in Fig. 7(right), we design an interactive real-time system for latent molecule manipulation, where the user can click random to randomly sample a molecule and freely select which model to interpret, which property to interpret, and tuning the slide bar manipulates the molecule accordingly in real-time. The demo video is anonymously provided at `https://drive.google.com/drive/folders/1N036p_5OfvGZybgPJ3Vw1ONXHVepimSR?usp=sharing`.

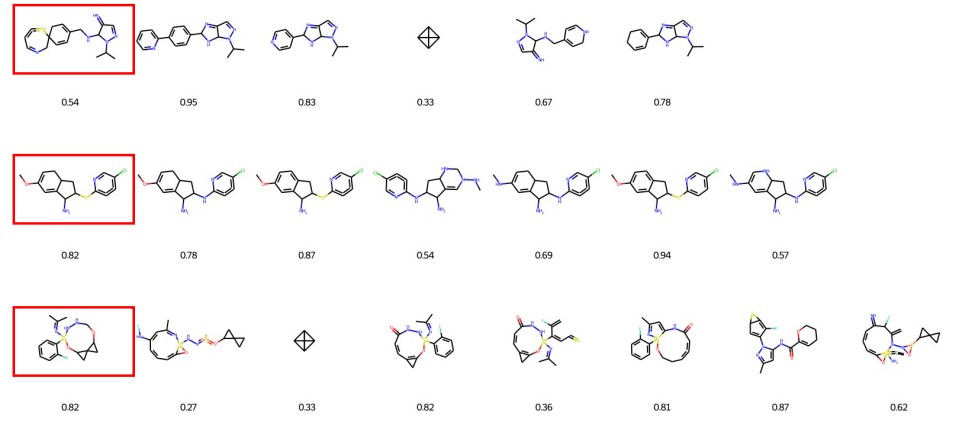

Figure 7: Optimizing molecular properties with optimization-based method.

Table 7: Single property molecule optimization for Penalized-logP and QED on the ZINC dataset

| Method | N | QED | pLogP |
|---|---|---|---|
| ZINC | Top 5 | $0.948 \pm 1.47e-4$ | $4.287 \pm 0.138$ |
| | Top 10 | $0.948 \pm 1.44e-4$ | $4.180 \pm 0.154$ |
| | Top 25 | $0.948 \pm 2.18e-4$ | $3.998 \pm 0.181$ |
| MoFlow | Top 5 | $0.948 \pm 1.26e-4$ | $3.972 \pm 0.075$ |
| | Top 10 | $0.948 \pm 1.38e-4$ | $3.879 \pm 0.110$ |
| | Top 25 | $0.948 \pm 2.37e-4$ | $3.766 \pm 0.118$ |
| ChemSpacE | Top 5 | $0.948 \pm 1.41e-4$ | $3.933 \pm 0.020$ |
| | Top 10 | $0.948 \pm 1.71e-4$ | $3.879 \pm 0.068$ |
| | Top 25 | $0.948 \pm 2.43e-4$ | $3.777 \pm 0.101$ |

Table 8: Constrained multi-property molecule optimization for Penalized-logP and QED on the ZINC dataset with two variants of gradient-based methods ($\delta$ is the threshold for similarity between the optimized and base molecules).

| MoFlow-1 | | | | | | |
|---|---|---|---|---|---|---|
| $\delta$ | QED Improvement | QED % Improvement | pLogP Improvement | pLogP % Improvement | Similarity | Success |
| **0.0** | $0.17 \pm 0.11$ | $42.06 \pm 35.69\%$ | $4.49 \pm 3.87$ | $51.00 \pm 29.36\%$ | $0.44 \pm 0.24$ | 91.50% |
| **0.2** | $0.16 \pm 0.11$ | $37.84 \pm 32.16\%$ | $4.42 \pm 3.78$ | $51.26 \pm 28.96\%$ | $0.48 \pm 0.21$ | 90.75% |
| **0.4** | $0.12 \pm 0.10$ | $29.53 \pm 27.45\%$ | $3.64 \pm 3.43$ | $44.61 \pm 29.34\%$ | $0.61 \pm 0.17$ | 73.25% |
| **0.6** | $0.07 \pm 0.08$ | $17.44 \pm 20.36\%$ | $1.85 \pm 2.18$ | $26.38 \pm 25.59\%$ | $0.78 \pm 0.15$ | 41.13% |
| MoFlow-2 | | | | | | |
| $\delta$ | QED Improvement | QED % Improvement | pLogP Improvement | pLogP % Improvement | Similarity | Success |
| **0.0** | $0.18 \pm 0.12$ | $45.09 \pm 39.71\%$ | $4.67 \pm 4.23$ | $50.74 \pm 28.79\%$ | $0.41 \pm 0.23$ | 92.88% |
| **0.2** | $0.16 \pm 0.11$ | $40.12 \pm 35.36\%$ | $4.48 \pm 3.78$ | $51.32 \pm 29.11\%$ | $0.47 \pm 0.20$ | 91.50% |
| **0.4** | $0.13 \pm 0.10$ | $31.25 \pm 29.87\%$ | $3.70 \pm 3.37$ | $45.16 \pm 29.27\%$ | $0.60 \pm 0.17$ | 74.88% |
| **0.6** | $0.07 \pm 0.08$ | $17.61 \pm 20.88\%$ | $1.97 \pm 2.51$ | $26.74 \pm 26.30\%$ | $0.78 \pm 0.15$ | 41.88% |
| **ChemSpacE** | | | | | | |
| $\delta$ | QED Improvement | QED % Improvement | pLogP Improvement | pLogP % Improvement | Similarity | Success |
| **0.0** | $0.20 \pm 0.12$ | $50.75 \pm 41.77\%$ | $4.66 \pm 4.34$ | $50.01 \pm 24.36\%$ | $0.34 \pm 0.23$ | 76.38% |
| **0.2** | $0.18 \pm 0.11$ | $42.70 \pm 32.87\%$ | $4.36 \pm 3.50$ | $51.57 \pm 28.27\%$ | $0.45 \pm 0.19$ | 76.75% |
| **0.4** | $0.14 \pm 0.10$ | $33.59 \pm 27.92\%$ | $3.78 \pm 3.49$ | $46.07 \pm 28.09\%$ | $0.58 \pm 0.16$ | 63.13% |
| **0.6** | $0.08 \pm 0.08$ | $20.12 \pm 22.33\%$ | $1.80 \pm 1.81$ | $26.77 \pm 24.75\%$ | $0.77 \pm 0.15$ | 32.13% |

Table 9: Quantitative Evaluation of Latent Manipulation.

| Datasets | Models | Validity (%) | Novelty (%) | Uniqueness (%) |
|---|---|---|---|---|
| QM9 | MoFlow | 100.00 | 98.23 | 98.27 |
| | Random | 91.60 | 91.60 | 8.06 |
| | Largest | 40.75 | 40.75 | 9.32 |
| | ChemSpacE | 91.63 | 88.71 | 24.23 |
| QM9 | HierVAE | 100.00 | 79.39 | 95.14 |
| | Random | 100.00 | 84.53 | 28.91 |
| | Largest | 100.00 | 84.05 | 27.26 |
| | ChemSpacE | 100.00 | 79.66 | 34.81 |
| ZINC | MoFlow | 100.00 | 100.00 | 100.00 |
| | Random | 69.98 | 69.98 | 29.04 |
| | Largest | 43.36 | 43.36 | 24.87 |
| | ChemSpacE | 71.26 | 71.26 | 15.82 |
| ChEMBL | HierVAE | 100.00 | 94.03 | 99.45 |
| | Random | 100.00 | 84.53 | 28.91 |
| | Largest | 100.00 | 93.00 | 55.09 |
| | ChemSpacE | 100.00 | 94.24 | 56.58 |

## E   Molecule Representations

**Molecule Graph.**   A molecule can be presented as a graph $X = (\mathcal{V}, \mathcal{E}, E, F)$, where $V$ denotes a set of $N$ vertices (*i.e.*, atoms), $\mathcal{E} \subseteq V \times V$ denotes a set of edges (*i.e.*, bonds), $F \in \{0,1\}^{N \times D}$ denotes the node features (*i.e.*, atom types) and $E \in \{0,1\}^{N \times N \times K}$ denotes the edge features (*i.e.*, bond types). The number of atom types and bond types are denoted by $D$ and $K$, respectively.

## F   Molecule Generative Models

In Table 10, we summarize a list of representative molecule generative models, which span various types of deep generative models, including the type of generative models, the type of generation process and whether latent space is learned. We also provide the formulation for two types of deep generative models (VAE and Flow) in Section F that are very popular for molecule generation task.

Table 10: A summary of mainstream molecule generative models.

| Prior Work | Generative Model | Sequential | Latent Space |
|---|---|---|---|
| JT-VAE (Jin et al., 2018a) | VAE | ✓ | ✓ |
| CGVAE (Liu et al., 2018) | VAE | ✓ | ✓ |
| MRNN (Popova et al., 2019) | RNN | ✓ | |
| GraphNVP (Madhawa et al., 2019) | Flow | | ✓ |
| GCPN (You et al., 2018a) | RL | ✓ | |
| GraphAF (Shi et al., 2020) | Flow | ✓ | |
| MoFlow (Zang & Wang, 2020) | Flow | | ✓ |
| HierVAE (Jin et al., 2020) | VAE | ✓ | ✓ |
| GraphEBM (Liu et al., 2021) | EBM | | |
| GraphDF (Luo et al., 2021) | Flow | ✓ | |

### F.1 Molecule Generative Model Formulation

**VAE.** gets a lower bound (ELBO) for the data log probability by introducing a proposal distribution.

$$
\begin{aligned}
\log p(x) &= \log \int_z p(x|z)p(z)dz \\
&\geq \log[\mathbb{E}_{q(z|x)}[p(x|z)] + \mathrm{KL}(q(z|x)||p(z))]
\end{aligned}
\tag{19}
$$

**Flow.** The key of Flow model is to design a invertible function with the following nice property:

$$
\begin{aligned}
z_0 &\sim p_0(z_0) \\
z_i &= f_i(z_{i-1}) \\
z_{i-1} &= f_i^{-1}(z_i) \\
p_i(z_i) &= p_{i-1}(z_{i-1})\left|\det\frac{df_i^{-1}}{dz_i}\right| = p_{i-1}(f_i^{-1}(z_i))\left|\det\frac{df_i^{-1}}{dz_i}\right|,
\end{aligned}
\tag{20}
$$

where $f_i$ is invertible function. To be more concrete, we can take $z_0$ as some tractable noise distribution, like Gaussian distribution, and repeating this for $K$ steps will lead to the data distribution, *i.e.*,:

$$
x = z_K = f_K \circ f_{K-1} \circ ... \circ f_1(z_0).
$$

Thus, the log likelihood of the data is as follows:

$$
\begin{aligned}
\log p(x) &= \log p_K(z_K) \\
&= \log p_{K-1}(z_{K-1}) - \log\left|\det\frac{df_K}{dz_{K-1}}\right| \\
&= \log p_{K-2}(z_{K-2}) - \log\left|\det\frac{df_{K-1}}{dz_{K-2}}\right| - \log\left|\det\frac{df_K}{dz_{K-1}}\right| \\
&= ... \\
&= \log p_0(z_0) - \sum_{i=1}^{K}\log\left|\det\frac{df_i}{dz_{i-1}}\right|
\end{aligned}
\tag{21}
$$

## G  Study of Molecular Properties

**List of Molecular Properties.** In total, study 208 molecular properties from the open chemistry library RDKit[2] and 4 important molecular properties including synthesis accessibility and binding affinity scores from TDC[3]. They are MaxEStateIndex, MinEStateIndex, MaxAbsEStateIndex, MinAbsEStateIndex, qed, MolWt, HeavyAtomMolWt, ExactMolWt, NumValenceElectrons, NumRadicalElectrons, MaxPartialCharge, MinPartialCharge, MaxAbsPartialCharge, MinAbsPartialCharge, FpDensityMorgan1, FpDensityMorgan2, FpDensityMorgan3, BCUT2D_MWHI, BCUT2D_MWLOW, BCUT2D_CHGHI, BCUT2D_CHGLO, BCUT2D_LOGPHI, BCUT2D_LOGPLOW, BCUT2D_MRHI, BCUT2D_MRLOW, BalabanJ, BertzCT, Chi0, Chi0n, Chi0v, Chi1, Chi1n, Chi1v, Chi2n, Chi2v, Chi3n, Chi3v, Chi4n, Chi4v, HallKierAlpha, Ipc, Kappa1, Kappa2, Kappa3, LabuteASA, PEOE_VSA1, PEOE_VSA10, PEOE_VSA11, PEOE_VSA12, PEOE_VSA13, PEOE_VSA14, PEOE_VSA2, PEOE_VSA3, PEOE_VSA4, PEOE_VSA5, PEOE_VSA6, PEOE_VSA7, PEOE_VSA8, PEOE_VSA9, SMR_VSA1, SMR_VSA10, SMR_VSA2, SMR_VSA3, SMR_VSA4, SMR_VSA5, SMR_VSA6, SMR_VSA7, SMR_VSA8, SMR_VSA9, SlogP_VSA1, SlogP_VSA10, SlogP_VSA11, SlogP_VSA12, SlogP_VSA2, SlogP_VSA3, SlogP_VSA4, SlogP_VSA5, SlogP_VSA6, SlogP_VSA7, SlogP_VSA8, SlogP_VSA9, TPSA, EState_VSA1, EState_VSA10, EState_VSA11, EState_VSA2, EState_VSA3, EState_VSA4, EState_VSA5, EState_VSA6, EState_VSA7, EState_VSA8, EState_VSA9, VSA_EState1,

---

[2]https://www.rdkit.org/docs/index.html
[3]https://tdcommons.ai/

VSA_EState10, VSA_EState2, VSA_EState3, VSA_EState4, VSA_EState5, VSA_EState6, VSA_EState7, VSA_EState8, VSA_EState9, FractionCSP3, HeavyAtomCount, NHOHCount, NOCount, NumAliphaticCarbocycles, NumAliphaticHeterocycles, NumAliphaticRings, NumAromaticCarbocycles, NumAromaticHeterocycles, NumAromaticRings, NumHAcceptors, NumHDonors, NumHeteroatoms, NumRotatableBonds, NumSaturatedCarbocycles, NumSaturatedHeterocycles, NumSaturatedRings, RingCount, MolLogP, MolMR, fr_Al_COO, fr_Al_OH, fr_Al_OH_noTert, fr_ArN, fr_Ar_COO, fr_Ar_N, fr_Ar_NH, fr_Ar_OH, fr_COO, fr_COO2, fr_C_O, fr_C_O_noCOO, fr_C_S, fr_HOCCN, fr_Imine, fr_NH0, fr_NH1, fr_NH2, fr_N_O, fr_Ndealkylation1, fr_Ndealkylation2, fr_Nhpyrrole, fr_SH, fr_aldehyde, fr_alkyl_carbamate, fr_alkyl_halide, fr_allylic_oxid, fr_amide, fr_amidine, fr_aniline, fr_aryl_methyl, fr_azide, fr_azo, fr_barbitur, fr_benzene, fr_benzodiazepine, fr_bicyclic, fr_diazo, fr_dihydropyridine, fr_epoxide, fr_ester, fr_ether, fr_furan, fr_guanido, fr_halogen, fr_hdrzine, fr_hdrzone, fr_imidazole, fr_imide, fr_isocyan, fr_isothiocyan, fr_ketone, fr_ketone_Topliss, fr_lactam, fr_lactone, fr_methoxy, fr_morpholine, fr_nitrile, fr_nitro, fr_nitro_arom, fr_nitro_arom_nonortho, fr_nitroso, fr_oxazole, fr_oxime, fr_para_hydroxylation, fr_phenol, fr_phenol_noOrthoHbond, fr_phos_acid, fr_phos_ester, fr_piperdine, fr_piperzine, fr_priamide, fr_prisulfonamd, fr_pyridine, fr_quatN, fr_sulfide, fr_sulfonamd, fr_sulfone, fr_term_acetylene, fr_tetrazole, fr_thiazole, fr_thiocyan, fr_thiophene, fr_unbrch_alkane, fr_urea, sa, drd2, jnk3, gsk3b.

However, not all of the molecular properties are varied in the three datasets. Specifically, **QM9** contains 29 frozen molecular properties, NumRadicalElectrons, SMR_VSA8, SlogP_VSA12, SlogP_VSA7, SlogP_VSA9, EState_VSA11, VSA_EState10, fr_C_S, fr_N_O, fr_SH, fr_azide, fr_azo, fr_barbitur, fr_benzodiazepine, fr_diazo, fr_hdrzine, fr_hdrzone, fr_isocyan, fr_isothiocyan, fr_nitroso, fr_phos_acid, fr_phos_ester, fr_prisulfonamd, fr_sulfide, fr_sulfonamd, fr_sulfone, fr_thiazole, fr_thiocyan, fr_thiophene, **ZINC** contains 4 frozen molecular properties, NumRadicalElectrons, SMR_VSA8, SlogP_VSA9, fr_prisulfonamd and **ChEMBL** contains only 3 frozen molecular properties, SMR_VSA8, SlogP_VSA9, fr_prisulfonamd.

**Inter-correlations of molecular properties.** In Fig. 8, we visualize the linear correlations between each pair of molecular properties across three datasets. From the heatmaps, we can observe that there are no linear correlations between half of the molecular properties, and similar patterns are observed in ZINC and ChEMBL datasets.

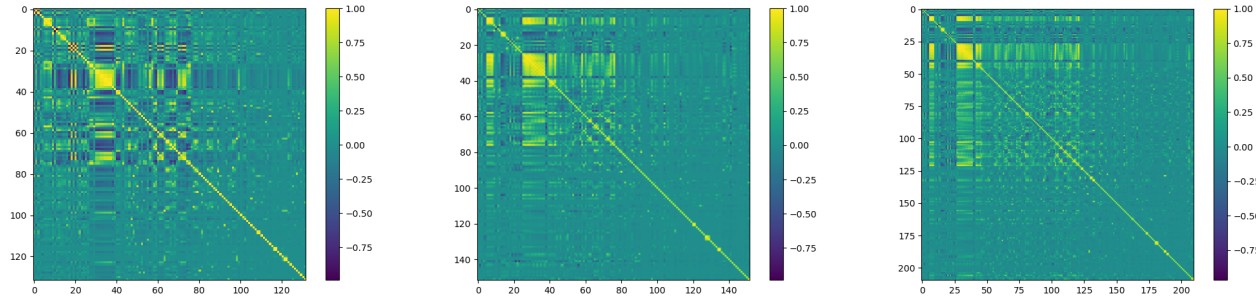

Figure 8: Inter-correlation heatmaps for studied molecular properties in QM9, ZINC and ChEMBL datasets.

**Molecular Property Distributions.** We visualize 7 molecular property distributions reported in section 4 in Fig. 9. From there, we can observe that the property distribution may vary a lot in terms of different datasets. Notably, the distributions of some properties, *e.g.*, QED, are very similar in ZINC and ChEMBL datasets, while some are quite different, *e.g.*, MolWt.

## H    Latent Space Evaluation

To evaluate the quality of the learned latent space, we utilize three disentanglement evaluation metrics, disentanglement, completeness and informativeness (Eastwood & Williams, 2018). To be specific, disentanglement measures the degree to which each latent dimension controls at most one molecular property, completeness

Figure 9: Property distributions of 7 randomly selected molecular properties on QM9, ZINC and ChEMBL datasets.

measures the degree to which each molecular property is governed by at most one latent dimension, and informativeness measures the prediction accuracy of molecular properties given the latent representation. From Table 11, we find MoFlow learns a better and more disentangled latent space than HierVAE. One possible reason is that MoFlow (369) has a larger latent space than HierVAE (32) since Flow restricts the latent size to be equal to the input size.

Table 11: Quantitative Evaluation of Disentanglement on Latent Space.

| Datasets | Models | Disentanglement | Completeness | Informativeness |
|---|---|---|---|---|
| QM9 | MoFlow | **0.24** | **0.57** | **0.83** |
| | HierVAE | 0.13 | 0.27 | 0.75 |
| ZINC | MoFlow | **0.40** | **0.62** | 0.87 |
| ChEMBL | HierVAE | **0.14** | **0.41** | **0.81** |

