# OpenReview forum: "ChemSpacE: Interpretable and Interactive Chemical Space Exploration"
_TMLR — Accepted by TMLR_

### Review · Reviewer_JCR3 · 2023-01-04

**Summary Of Contributions:**

This paper proposes using existing latent variable generative models of chemicals along with linear latent space interpolations to manipulate chemicals for particular properties.

It then evaluates how well this performs compared to some simple baselines.

**Audience:**

No

**Claims And Evidence:**

No

**Requested Changes:**

The discussion in section "Molecular Generative Models" is incorrect. I don't think its fair to say that the encoding and decoding process for a VAE are described by deterministic maps as shown in Equation 1.  I feel as though that sort of misses the point that a VAE is a probabilistic model with a stochastic encoder / decoder.  The encoding process maps an input $x$ to a distribution in the latent space $p(z|x)$.  Granted, for the simplest of VAEs which use mean field gaussian encoders, potentially with fixed variance this map is characterized by the map for the means alone which I think is what is meant to be invoked here.

In Table 3, e-notation floating point values were dumped into the LaTeX table, i.e. "2.75e-2" which turns into $2.75e-2$ and looks terrible.  Please use $2.75\times 10^{-2}$ instead, as is standard.

The discussion in Section 3.2 isn't the most clear.  I'd suggest reframing the discussion of your method in the context of a probabilistic model to better align with the audience.

**Strengths And Weaknesses:**

Overall, I don't think this paper is a good fit for TMLR.  I don't think it's a great fit for the audience.

I don't mean to sound harsh but I don't believe there is a lot of machine learning research content in this work.  The proposed method is a linear latent space interpolation. Latent space interpolations are well known and linear latent interpolations are usually a first sort of sanity check done to ensure a latent variable model is doing something interesting.  In fact, one of the models used in this work, MoFlow, had in their Figure 4 (with essentially the same visual styling as Figure 2 here) a visualization of a linear latent space interpolation.

Perhaps it is news to the chemical modelling community just how effective linear latent space interpolations can be for manipulating chemical properties, but then the proper audience should be more focused group of chemical modelers.

If you were looking to expand the research, into something more appropriate for a TMLR audience, I'd consider exploring a wider class of semi-supervised or conditional generative models of molecules that take into account property information.  That is, in this work, first a latent variable model is built, which we can view a a joint model over both chemicals and some latent: $p(z,x)$ and then after the fact, some model is learned to try to associate or align chemical properties with the learned latent, $p(p|z)$, which for this work a simple linear model was used.  First you could consider much richer models for $p(p|z)$ beyond simpler linear ones.  But more generally, I think its much more interesting to ask whether chemical properties can lead to better, more structured generative models.  Instead of bolting the property identification on at the end and hoping that the learned latent variable model has distentangled the desired chemical properties, why not model them as well?  Learn a joint $p(z,x,p)$ model directly.  There are lots of interesting directions to consider along this path, including many standard approaches like semi-supervised VAEs (arxiv/1406.5298).

---

### Review · Reviewer_4w7v · 2023-01-25

**Summary Of Contributions:**

The work presents an interactive tool to navigate the latent space of a pre-trained generative model, in the applied domain of molecule design and discovery. The particular technique of traversing a pre-trained latent space is not particularly novel, but the application to  molecule manipulation (a new task defined by this work) is useful and interesting. The tool can be used to understand and explore the design space, and when combined with human-in-the-loop, optimize molecule design. Experiment results are compelling. Some technical details are missing or buried in Appendix. The paper could be strengthened by adding those details back.





**Audience:**

Yes

**Claims And Evidence:**

Yes

**Requested Changes:**

The two proposed metrics, strict success rate (SSR) and relaxed success rate (RSR), are rather important concepts, but their equations are buried in appendix. The authors should consider moving them to the main paper, especially when they are used to evaluate the proposed method in multiple tables later on.

A few citations are wrongly formatted. For example, "...during the searching or optimization process Huang et al. (2021)." should be "...during the searching or optimization process (Huang et al., 2021)." Also: "...emergent properties of the latent space learned by molecule generative models Gómez-Bombarelli et al. (2018); Zang & Wang (2020)"

Gradient-based method is an important baseline, but is given no explanation how it works.

In Table 1, it is confusing to see new combined names like "MoFlow-R" and "HierVAE-C". Consider using two separate columns to indicate the generative model and the latent exploration method.

**Strengths And Weaknesses:**

Strengths:
 - The paper is well written, and easy to follow
 - The work addresses an important application domain
 - It proposes a new task, new metrics, as well as a new method
 - The interactive demo is well designed

Weaknesses:
 - Molecule manipulation baselines are rather weak, especially "Random manipulation" and "Largest range manipulation"
 - "Gradient-based method" as a baseline was not given enough details
 - a few formatting mistakes for citations

---

> ### Author Response · Authors · 2023-01-29
> **Response to Reviewer 4w7v**
>
> We thank the reviewer for the valuable comments and suggestions, we update the manuscript with changes marked in blue and provide a detailed response to each point raised below:
>
> Q1: Molecule manipulation baselines are rather weak, especially "Random manipulation" and "Largest range manipulation"
>
> R1: Thanks for the comment. They are indeed weak since the random manipulation even requires no data and the largest range just takes the molecules with the largest property value and the smallest property value as a simple heuristic. The gradient-based baseline is much stronger.
>
> Q2: "Gradient-based method" as a baseline was not given enough details
>
> R2: Thanks for the comment. We have added detailed explanations in Section 4.1.
>
> “Gradient-based method optimizes the molecular property of the generated molecules by searching a latent vector with the optimized molecular property via gradient ascent/descent. Specifically, it requires pre-training a property predictor on the latent space. It first initializes a latent vector and optimizes the latent vector to maximize/minimize the output of the predicted property value.”
>
> Q3: a few formatting mistakes for citations
>
> R3: Thanks for the comment. We have fixed them.
>
> Q4: In Table 1, it is confusing to see new combined names like "MoFlow-R" and "HierVAE-C". Consider using two separate columns to indicate the generative model and the latent exploration method.
>
> R4: Thanks for the comment. We have fixed Table 1 and 6 accordingly.
>
> Q5: The authors should consider moving them to the main paper, especially when they are used to evaluate the proposed method in multiple tables later on.
>
> R5: Thanks for the comment. We have moved them back to Section 2.

---

### Review · Reviewer_fKVq · 2023-01-30

**Summary Of Contributions:**

Authors describe a simple methodology called ChemSpacE to generate plausible molecules that demonstrate a target property by moving towards a decision boundary explicitly computed for that property, in an already existing latent space of a pre-trained deep generative model. As one can keep track of the molecular changes along this path, the method provides outcomes interpretable by humans. In effect, the model is analogous to training a linear support vector machine for the target property in the latent space of the generative model, after which one can move along the hyperplane normal towards the desired property. Authors design evaluation criteria and benchmark the performance of ChemSpacE in manipulating molecules using various public datasets and models.

**Audience:**

Yes

**Broader Impact Concerns:**

The reviewer could not identify any concerns on ethical implications of the work.

**Claims And Evidence:**

Yes

**Requested Changes:**

[Critical]:
- Please describe the gradient descent method used and discuss its suitability for serving as a fair baseline.
- Given that one of the main contributions is "demonstrate effectiveness and efficiency of the authors' method in molecule optimization ..." please, provide or at least comment on an alternative global or Bayesian optimization approach that would be more fair to compare with the presented method.

[Not critical]:
- Please provide a brief description of target properties QED, logP etc.
- Figure and Table numbers appear out of order. Please check.
- What does k denote in Eq.8?
- What are the functions L and G in Eqs. (13) to (18)?
- Please leave more space between subfigures in Figure 2 so the readers can more clearly see what visual elements belong to which subfigure, and also explain how Fig2(c) is generated.
- Please elaborate how Eq.9 helps find direction in multiple property scenarios.
- Please clarify how ChemSpacE “can be incorporated in various pre-trained state-of-the-art molecule generative models without retraining”. Would the method not require finding the decision boundary for a property in each new generative model’s latent space?
- What do the “manipulation ranges” mean as listed in the Hyperparameters section? Please explain how these should be interpreted.


**Strengths And Weaknesses:**

*Strengths:* The method addresses a specialized task designed by the authors; i.e. manipulating a starting molecule step by step towards a certain property in an interpretable way. An underlying assumption is the linear separability of the generator's latent space to target property classes. This assumption makes the methodology more suitable for human-in-the-loop exploration but still leads to a competitive optimization strategy. Authors provide means for users to try the manipulation method interactively.

*Weaknesses:* There is an emphasis on how ChemSpacE outperforms two simple baselines (random manipulation and largest range manipulation) and gradient descent/ascend in terms of success rate, speed up, data efficiency and so on. The first two baselines would be lower bounds for any working method for the task as defined by the authors. However, it is unclear what the “gradient descent” based method entails in the manuscript, and if it is fair to compare a local minima finding strategy based on gradients to the author's traversing of the entire latent space in linear strides. Performance is measured  using a continuous metric (success rate as defined by authors) that penalizes the former and favors the latter. Gradient descent would get stuck when initialized in local minima hence perform poorly on the defined metric, whereas authors’ method isn’t constrained by that; in other words, the objectives of the algorithms are different. I think the author's method is still useful, but baselines are not as informative as presented.

---

> ### Author Response · Authors · 2023-01-31
> **Response to Reviewer fKVq (1/2)**
>
> We thank the reviewer for the valuable comments and suggestions, we update the manuscript with changes marked in blue and provide a detailed response to each point raised below:
>
> Q1: Please describe the gradient descent method used and discuss its suitability for serving as a fair baseline.
>
> R1: Thanks for the comment. We have added detailed explanations in Section 4.1. We compare our method with the gradient-based method because it was a popular choice in the literature [1, 2]. Despite gradient-based methods may suffer from local directions, it serves as a good baseline for molecule optimization where the main goal is to optimize the molecular properties (which our method is on par with them). We also evaluate them on the molecule manipulation task to show that our method could lead to more interpretable and smooth manipulation length leveraging the observation about the latent space of molecular generative models.
>
> “Gradient-based method optimizes the molecular property of the generated molecules by searching a latent vector with the optimized molecular property via gradient ascent/descent. Specifically, it requires pre-training a property predictor on the latent space. It first initializes a latent vector and optimizes the latent vector to maximize/minimize the output of the predicted property value.”
>
>
> Q2: Given that one of the main contributions is "demonstrate effectiveness and efficiency of the authors' method in molecule optimization ..." please, provide or at least comment on an alternative global or Bayesian optimization approach that would be more fair to compare with the presented method.
>
> R2: Thanks for the comment. Despite Bayesian optimization and other global optimization methods being suitable for the task, our main goal is not to beat the other methods. Our main goal is to leverage a very efficient (linear model) yet effective method, shown to be compared to the commonly used gradient-based methods. We bring in another perspective about the interpretability and interactive design for molecules. We will leave as future work to explore the latent space of molecular generative models.
>
> Q3: Please provide a brief description of target properties QED, logP etc.
>
> R3: Thanks for the comment. We have added the explanation in the experiment set-up.
>
> “QED is a quantitative estimate of drug-likeness. PLogP refers to the partition coefficient logarithm of octanol-water which measures the lipophilicity and water solubility. SA denotes the synthesis accessibility score. MolWt denotes the molecular weight. DRD2, JNK3 and GSK3B are three binding affinity scores.”
>
>
> Q4: Figure and Table numbers appear out of order. Please check.
>
> R4: Thanks for the comment. We have adjusted the text, tables and figures.
>
> Q5: What does k denote in Eq.8?
>
> R5: Thanks for the comment. We have clarified $k$ is a scale factor such that the property might be a factor $k$ of the step number $\alpha$.
>
> Q6: What are the functions L and G in Eqs. (13) to (18)?
>
> R6: Thanks for the comment. L and G are two versions for evaluating relaxed success rates, since we need to set the tolerance variable $\epsilon$, we need to determine whether it is local (depending on the property range of the current path) or global (depending on the property range of the dataset)
>
> “We also provide two implementations of relaxed success rate, which defines different tolerance variables ε with local relaxed constraint (RSR-L) and global relaxed constraint (RSR-G)”
>
> Q7: Please leave more space between subfigures in Figure 2 so the readers can more clearly see what visual elements belong to which subfigure, and also explain how Fig2(c) is generated.
>
> R7: Thanks for the comment. We have fixed it and added the explanation.
>
> “Latent property boundary is visualized for QED property for MoFlow trained on ZINC by reducing the dimension of the latent vectors by PCA”
>
> Q8: Please elaborate how Eq.9 helps find direction in multiple property scenarios.
>
> R8: Thanks for the comment. We identify all the directions in the same way by leveraging a linear model to find the separation boundary of the molecular property in the latent space. However, for multi-property scenarios, as different molecular properties may correlate with each other (positively or negatively), we take only their positively correlated directions to manipulate each property rather than move the molecules onto the mean of the two directions.

---

> > ### Author Response · Authors · 2023-01-31
> > **Response to Reviewer fKVq (2/2)**
> >
> >
> >
> >
> > Q9: Please clarify how ChemSpacE “can be incorporated in various pre-trained state-of-the-art molecule generative models without retraining”. Would the method not require finding the decision boundary for a property in each new generative model’s latent space?
> >
> > R9: Thanks for the comment. Sorry for the confusion. We meant the re-training of the generative model is not required. In fact, this is very important for efficiency, e.g. some methods achieve controllable generation by training a class-conditioned generative model. To clarify, we have fixed the text:
> >
> > “We develop an efficient model-agnostic method named \emph{ChemSpacE} for molecule manipulation, which can be incorporated in various pre-trained state-of-the-art molecule generative models without re-training or modifying the pre-trained generative models”
> >
> > Q10: What do the “manipulation ranges” mean as listed in the Hyperparameters section? Please explain how these should be interpreted.
> >
> > R10: Thanks for the comment. The manipulation range is in fact the most important hyperparameter as it controls how far we travel in the latent space. It is intuitive that when it is small, we are in the high density region with higher validity for the generated molecules, but often less attractive properties. When we aim to optimize molecular property, we may need to go with a larger range to travel to the low density region which validity of the generated molecules may be lower.
> >
> > [1] Zang, C. and Wang, F., 2020, August. MoFlow: an invertible flow model for generating molecular graphs. In Proceedings of the 26th ACM SIGKDD International Conference on Knowledge Discovery & Data Mining (pp. 617-626).
> >
> > [2] Liu, Q., Allamanis, M., Brockschmidt, M. and Gaunt, A., 2018. Constrained graph variational autoencoders for molecule design. Advances in neural information processing systems, 31.

---

### Author Response · Authors · 2023-02-23
**Thanks for the comments!**

Dear reviewers,

We appreciate your valuable comments which have helped us greatly improve the manuscript! We are happy to discuss/address any further concerns you may have.

Thanks again for your time and effort in reviewing our work!

---

### Decision · Action_Editors · 2023-03-11

**Recommendation:** Accept as is

**Comment:**

The reviewers were concerned about the level of machine learning innovation in the paper, however agreed after the rebuttal period that claims were supported by evidence. Authors successfully argued that while linear interpolations in feature space are common, their approach to identifying interesting directions in the feature space was novel and of interest to the community.

**Audience:**

The official evaluation criteria asks "Would at least some individuals in TMLR's audience be interested in knowing the findings of this paper?". AI for science is a growing subfield with strong interest both from the ML community and from the natural sciences community. As the reviewers stated, some individuals of the TMLR audience would be interested in the findings of this paper.

**Claims And Evidence:**

- All reviewers were leaning towards acceptance at the end of the rebuttal period.
- Authors successfully addressed the reviewers concerns.
- At the end of the rebuttal period, reviewers were convinced that the claims in the paper are supported by evidence. However, two of the reviewers still had a concern that the paper was not necessarily of interest to the whole TMLR community, and might be a better fit for a specialized chemistry journal.